# Anti-Influenza A Potential of *Tagetes erecta* Linn. Extract Based on Bioinformatics Analysis and In Vitro Assays

**DOI:** 10.3390/ijms25137065

**Published:** 2024-06-27

**Authors:** Minjee Kim, Aleksandra Nowakowska, Jaebum Kim, Young Bong Kim

**Affiliations:** Department of Biomedical Science and Engineering, Konkuk University, Seoul 05029, Republic of Korea; mj0411@konkuk.ac.kr (M.K.); szalonystudent@gmail.com (A.N.); jbkim@konkuk.ac.kr (J.K.)

**Keywords:** network pharmacology, molecular docking, influenza, antiviral treatment, *Tagetes erecta*, bioinformatics

## Abstract

*Tagetes erecta* Linn. (TE) is traditionally used to treat cardiovascular, renal, and gastrointestinal diseases. In this study, we investigated the active compounds and targets of TE extract that may exert antiviral effects against influenza A. Active compounds and targets of TE extract were identified using the Traditional Chinese Medicine Systems Pharmacology database (TCSMP). The influenza A-related gene set was screened using GeneCards and the Kyoto Encyclopedia of Genes and Genomes (KEGG). A protein–protein interaction (PPI) network was built to establish the hub targets. Pathway and target studies were conducted using Gene Expression Omnibus (GEO). The interactions between active compounds and potential targets were assessed by molecular docking. An in vitro study was performed using antiviral and plaque reduction assays. From the compound and target search, we identified 6 active compounds and 95 potential targets. We retrieved 887 influenza-associated target genes and determined 14 intersecting core targets between TE and influenza. After constructing a compound–target network, we discovered lutein and beta-carotene to be the key compounds. Next, PPI network analysis identified the top three hub genes associated with influenza (IL-6, HIF1A, and IL-1β). Similarly, GEO analysis revealed IL-6, TGFB1, and CXCL8 to be the top three target genes. In our docking study, we identified that lutein and IL-6 had the strongest bindings. Our in vitro experimental results revealed that the TE extract exhibited therapeutic rather than prophylactic effects on influenza disease. We identified lutein as a main active compound in TE extract, and IL-6 as an important target associated with influenza, by using data mining and bioinformatics. Our in vitro findings indicated that TE extract exerted protective properties against the influenza A virus. We speculated that lutein, as a key active component in TE extract, is largely responsible for its antiviral effects. Therefore, we suggest TE extract as an alternative in the treatment of influenza.

## 1. Introduction

Influenza is a respiratory disease that occurs due to continuous antigenic drift and sporadic shift [1,2]. The influenza virus has a negative-sense RNA genome without a proofreading mechanism, leading to constant mutations over the years. It is reported that vaccines are the first line of defense against influenza; therefore, sequence analyses are commonly used to develop effective global vaccines [1]. Therapeutic interventions are also important, particularly when vaccines are unavailable or ineffective. However, to date, only one class of drugs has been approved for antiviral treatment. These are neuraminidase (NA) inhibitors (NAIs), virus-specific drugs that bind to the active site of the virus. However, a major drawback of NAIs is that they must be administered within 48 h of onset, a challenging requirement in many countries. Moreover, the NAI-resistant strains lower the efficacy of available antiviral agents, necessitating the development of novel influenza drugs [3].

Natural products are commonly used in medicine because they contain bioactive compounds that control cellular targets in many diseases [4]. Their main benefits include few adverse effects, affordability, and accessibility. Currently, high-throughput screening is the primary technique used to evaluate the pharmacological effects of natural products with potential to be used as herbal medicines [5]. However, with the rapidly increasing need for new drugs, the traditional drug-discovery strategy of “one drug–one target–one disease” has become inefficient and time-consuming [6]. Instead, the concept of “network pharmacology” has emerged, the idea of developing multi-target drugs for complex diseases by predicting genes controlled by bioactive compounds [7].

*Tagetes erecta* Linn. (TE; family Asteraceae) exerts several biological benefits, including antioxidant, antiproliferative, and antidiabetic effects [8]. In South America, TE flower extracts were commonly used to treat cardiovascular, renal, and gastrointestinal diseases [9]. Research in rodents suggests that TE flower extracts reduce inflammation and improve gastric diseases [10]. TE extracts contain high levels of lutein that are used as nutritional supplements to prevent aging or disease-related loss of visual acuity [11].

In this study, we aimed to determine the action of TE extract against influenza infection, in addition to identifying the major associated compounds, targets, and associated pathways. First, we constructed a compound–target network to identify active compounds, their targets, and pathways associated with the influenza virus. Second, we performed molecular docking analysis to identify the potential compound and target. Finally, we evaluated in vitro cytotoxicity and antiviral effects of the TE extract.

## 2. Results

### 2.1. In Silico Findings

#### 2.1.1. Active Compounds and Target Prediction

We selected six active compounds from TCMSP based on the criteria of OB ≥ 20 and DL ≥ 0.1, following the database suggestions (Table 1). The plant extract composition is given in Appendix A. The targets regulated by these compounds in the TE extract were queried against the TCMSP and NCBI databases to obtain 95 targets.

#### 2.1.2. Influenza Target Prediction

We obtained 160 and 720 targets with the search terms “influenza” and “Homo sapiens” from KEGG and GeneCards, respectively. Removing duplicates yielded 14 intersecting target genes associated with both the TE extract and influenza. These 14 targets were further analyzed using a compound–disease target network and four compounds (lutein, beta-carotene, beta-sitosterol, and alpha-carotene). The results indicated that lutein had the highest degree of connection, followed by beta-carotene (Figure 1).

#### 2.1.3. Construction of the Protein–Protein Interaction (PPI) Network of Potential Targets

We analyzed 14 intersection targets of TE–active compounds associated with influenza using the Search Tool for the Retrieval of Interacting Genes/Proteins (STRING) database (https://string-db.org/ accessed on 2 February 2024) and imported them into Cytoscape to construct a compound–target network (Figure 2).

The network contained 14 nodes linked to 49 edges (see Table 2 for targets). The PPI network analysis revealed that IL-6 exhibited the highest degree of connectivity.

#### 2.1.4. GEO Analysis

We obtained 20,396 H1N1 influenza-related disease genes from the GSE48466 dataset, then characterized the expression data of seven overlapping targets from GEO case samples. Volcano and box plots of the gene expression data are given in Appendix A. A gene-difference heatmap (Figure 3) revealed that the average expression levels of H1N1 patients compared to healthy controls were ranked as IL6 > TGFB1 > CXCL8 > RELA > TNFRSF10B > BAX > ATM. Details of the expression levels are given in Appendix A.

#### 2.1.5. KEGG Pathway and GO Analysis

The KEGG analysis revealed 81 enriched signaling pathways, and the top 10 pathway enrichment rankings were plotted in bubble diagrams (Figure 4). Influenza was the disease with the third highest correlation after diabetic complications and non-alcoholic fatty liver disease.

For GO biological process (BP) analysis, we used seven targets to draw a chord diagram containing four biological regulatory processes (Figure 5). Six targets were enriched in a GO chord. IL6, TGFB1, CXCL8, and RELA were associated with inflammatory response (GO:0006954), while IL6, TGFB1, and BAX were involved in cellular response to the virus (GO:0098586).

#### 2.1.6. Molecular Docking of Active Compounds with Hub Target Genes

We conducted molecular docking to evaluate the potential target bindings with lutein and beta-carotene in the TE extract (Table 3). Three main targets retrieved from GEO (IL-6, TGFB1, and CXCL8) were docked to measure the binding affinity with the two compounds. IL-6 exhibited the strongest binding affinity of the three targets with lutein and beta-carotene. Between the two compounds, lutein bound more strongly to IL-6, whereas beta-carotene bound more strongly to TFGB1 and CXCL8. We then investigated the 2D binding of lutein and IL-6 in Biovia Discovery Studio (Figure 6). The key bonds were alkyl bonds with LYS66, PHE74, CYS73, ALA180, and PHE78 as well as pi-alkyl bonds with PHE74 and ARG179.

### 2.2. In Vitro Findings

#### 2.2.1. Cytotoxicity and Antiviral Activity of TE Extract

A cytotoxicity test was conducted to determine the dose and toxicity at the cell level (Appendix A). The cell viability levels with TE extract at the lowest (500 µg/mL) to highest concentration (2000 µg/mL) were between 60–95%. The mean cytotoxicity concentration of the 50% (CC50) value was higher than 2000 µg/mL (Table 4). The IC50 value was defined based on a post-treatment assay. The IC50 value was higher than 70 µg/mL and the SI value of >30. Four separate experiments were performed to determine whether the TE extract exhibits anti-influenza activity at different stages of the viral life cycle (Figure 7). Our in vitro results revealed that the TE extract had effects in the attachment inhibition and co-treatment assays (Figure 7B,C). In both cases, the extract at doses of 1000 and 500 µL/mL significantly improved cell viability from untreated conditions. However, antiviral activity was absent in the pre-treatment assay (Figure 7A), indicating that TE extract is unlikely to elicit protective effects against influenza infection. Instead, the extract exhibited antiviral properties later in infection. Post-treatment yielded the best results among the four assays, with cell viability increasing to 31 µg/mL. Interestingly, incubation of the extract with infected cells inhibited viral activity and increased cell growth, leading to cell viability of >100% from the control levels.

#### 2.2.2. Time-of-Addition Assay

To further investigate the therapeutic effect of the TE extract in a post-treatment test and to note the stage of viral life cycle inhibition, a time-of-additional assay was conducted. The results showed a decreasing trend with delayed treatment initiation (Figure 8), suggesting that the best antiviral effects are assured with treatment administered shortly after (2 h) infection.

#### 2.2.3. Plaque Reduction Assay

The antiviral activity of the TE extract was confirmed using a plaque reduction assay (Figure 9), comparing the number of plaques in treated infected (50 PFU/well) cells to the number in untreated infected (50 PFU/well) cells. Compared with the infection-only control group, the plaque count decreased to approximately 60% post-treatment with 1000 µL/mL of the TE extract and stayed at approximately 70% for other groups.

## 3. Discussion

Influenza is a serious threat to public health [12]. Due to antigenic shift and the complications of influenza infection, influenza vaccines are formulated annually to match the circulating strains [13]. Antiviral agents that target influenza have been developed, but the use of these agents have many limitations [2].

Although TE was traditionally used to treat gastrointestinal diseases, and is considered to have medicinal properties such as anthelmintic, diuretic, and sedative effects, there are no studies on the effectiveness of TE against influenza virus [10]. In this study, we aimed to find active compounds and targets of TE associated with influenza for the first time by using data mining, bioinformatics, and systems biology. We identified two core active compounds—lutein and beta-carotene—via compound–disease target network analysis. For the target search, we applied a PPI network to identify the top three hub genes associated with influenza (IL-6, HIF1A, and IL-1β). From our PPI analysis, we identified HIF1A as one of the key targets associated with influenza. HIF1A is a key mediator in inflammation and is involved in the transcriptional regulation of cytokines, such as IL-6 and TNF-α [14]. Recent research reported that influenza infection induced the nuclear translocation of HIF1A, which may promote the production of proinflammatory cytokines [15]. Influenza infection also induces IL-1β, resulting in lung inflammation [16]. It was reported that significantly higher levels of IL-1β and IL-6 were detected in influenza patients with more severe conditions [17]. Similarly, GEO analysis revealed IL-6, TGFB1, and CXCL8 to be the top three target genes. TGF-beta is reported to control the immune responses—both inflammatory and regulatory [18]. Additionally, influenza infection stimulated CXCL8 secretion by primary human alveolar epithelial cells [19]. In influenza, disease severity is strongly associated with high levels of circulating IL-6-induced proteins, and monocyte chemotactic protein [20,21]. In our study, we identified IL-6 as a hub target in both PPI and GEO analysis. IL-6 levels were reported to be significantly increased in the sera of patients with uncomplicated influenza [20]. Moreover, elevated levels of the pro-inflammatory cytokine IL-6 are correlated with a high number of hospital admissions [22]. These findings correspond to our study result on potential targets associated with influenza infection. Herein, we conducted molecular docking to assess target binding of the two main compounds in TE extract. IL-6 exhibited the strongest binding affinity among the three selected targets with lutein and beta-carotene. We found that lutein binds more strongly than beta-carotene to IL-6. We speculated that lutein, as a key component in TE extract, is largely responsible for its antiviral effects by possibly reducing IL-6 level.

Virus invasion causes oxidative stress, resulting in inflammation and an exaggerated immune response, also known as cytokine storm [23]. Antioxidant therapy in combination with antiviral drugs has been suggested to reduce the lethal effects of influenza infection [24]. Our two main compounds, lutein and beta-carotene, belong to a carotenoid family that comprises natural lipid-soluble antioxidants. We speculate that the antioxidant potential of lutein and beta-carotene may include a reduction in inflammatory mediators and the prevention of viral mutations [25]. Antioxidants represent a potential therapeutic option to prevent influenza infection [26]. Therefore, the development of novel antioxidant compounds for influenza virus research and clinical use is needed to reinforce their potential.

Finally, we validated the anti-influenza effects of TE extract in MDCK cells and assessed its effects on the viral cycle. The post-treatment assay yielded the most potent effects and was effective up to a concentration of 31 µg/mL; this value was very low compared to the concentrations obtained in the other three assays. The time-of-addition assay revealed that the treatment applied shortly after infection (2 h) was most effective. Moreover, infected cell growth increased under incubation with TE extract, an improvement that may be attributable to the antioxidant activity of the natural compounds we identified.

With the continued advancement of big data, it is possible to screen and develop drugs from natural products using computer science [27]. Bioinformatics-driven drug screenings are cost- and time-effective research tools in developing useful pharmacological agents [28]. The primary focus of this study was to retrieve a list of active compounds of TE and target proteins of influenza infection through extensive data mining. The novelty of this study is that it is the first study of the effect of TE extract on influenza infection, and it was mainly conducted with computational work. The results of this study differ from the previously known effects of TE extract. However, further studies involving oxidative stress with influenza and its underlying mechanism will be required to provide a therapeutic option for the prevention and the control of influenza virus infection.

## 4. Materials and Methods

### 4.1. In Silico Analyses

#### 4.1.1. Identification of Potential Active Compounds in TE Extract

Known TE compounds were searched in Dr. Duke’s Phytochemical and Ethnobotanical database (version 1.10.07; https://phytochem.nal.usda.gov/phytochem accessed on 2 February 2024) and the Traditional Chinese Medicine Systems Pharmacology (TCMSP) database (version 2.3; http://tcmspw.com/tcmsp.php accessed on 2 February 2024). Based on the oral bioavailability (OB) and drug-likeness (DL) results from TCMSP, active compounds were then identified with an in silico absorption, distribution, metabolism, and excretion screening model.

#### 4.1.2. Potential Targets of TE Extract and Influenza Virus

Regulatory target genes associated with the identified TE compounds were investigated using TCMSP and DrugBank (version 5.1.10; https://www.drugbank.ca accessed on 2 February 2024). Influenza virus-associated target genes (Homo sapiens) were obtained from Kyoto Encyclopedia of Genes and Genomes (KEGG) and GeneCards (version 5.14.0; https://www.genecards.org/ accessed on 2 February 2024).

#### 4.1.3. Network Construction and Pathway Analysis

Compounds and intersecting core target networks were constructed using Cytoscape (version 3.9.0), and then analyzed to determine hub compounds and targets. Gene Ontology (GO) and pathway analyses of predicted targets were performed with the Database for Annotation, Visualization, and Integrated Discovery (DAVID, https://david.ncifcrf.gov accessed on 4 February 2024) and Kyoto Encyclopedia of Genes and Genomes (KEGG) [29], respectively.

#### 4.1.4. Gene Difference and Gene Enrichment Analyses

The Gene Expression Omnibus (GEO, https://www.ncbi.nlm.nih.gov/geo/ accessed on 6 February 2024) from NCBI was used to further screen influenza targets from clinical data. For preprocessing and normalization, we used the GEO2R analysis tool provided by NCBI GEO. Influenza clinical case samples were retrieved from the GSE48466 dataset, containing twelve clinical samples (nine from patients infected with human influenza H1N1 and three from healthy controls). An analysis of gene differences across GSE clinical samples allowed for the construction of an intuitive heatmap that further improved target screening.

#### 4.1.5. Molecular Docking Analysis

The major compounds in the TE extract were docked into three main targets retrieved from the network analysis to understand target specificity. The three-dimensional (3D) structures of the active compounds (lutein and beta-carotene) were downloaded from the PubChem database (https://pubchem.ncbi.nlm.nih.gov/ accessed on 24 February 2024) and converted to PDB files using Biovia Discovery Studio (version 20.1.0.19295). The 3D structures of the main targets were retrieved from the Research Collaboratory for Structural Bioinformatics (RCSB) Protein Data Bank (for the targets, IL-6 [PDB ID: 1P9M], transforming growth factor beta 1 [TGFB1; PDB ID: 5VQP], and IL-8 [CXCL8, PDB ID: 3IL8]). All the targets were converted to PDB files using Biovia Discovery Studio (ver. 4.5). Ligand-target dockings were conducted with AutoDock Vina (ver. 1.1.2) using PyRx (ver. 0.9.6) based on a scoring system. The CASTp (Computer Atlas of Surface Topology of proteins) server (ver. 3.0) was used to locate the active sites of the following targets: IL-6, TGFB1, and IL-6. Two-dimensional ligand–target interaction was performed using Biovia Discovery Studio.

### 4.2. In Vitro Experiment

#### 4.2.1. Plant Material and Extraction

TE flowers were grown at Woori Bio Smart farm (Gyeongsan, Republic of Korea). The plant materials (5 g) were ground and extracted using 100% ethanol (100 mL, reflux extraction for 3 h). Ethanol was used as it is safe for infused edibles and provides consistent results [30]. The extracts were filtered and evaporated at a low temperature under reduced pressure. After the extraction, the sample (450 mg) was provided by Woori Bio for the in vitro experiment. This research on plants complies with relevant institutional, national, and international guidelines and legislation. The extracted material has been deposited at Konkuk University (Seoul, Republic of Korea) and Woori Bio Smart farm.

#### 4.2.2. Cells and Viral Infection

Madin–Darby canine kidney (MDCK) cells were obtained from the American Type Culture Collection (ATCC, Manassas, VA, USA) and cultured in Eagle’s minimum essential medium (EMEM; ATCC) containing 10% fetal bovine serum (Gibco, Carlsbad, CA, USA) and 1% penicillin/streptomycin at 37 °C and 5% CO_2_. Human influenza type A/California/07/09 (H1N1) was provided by the Centers for Disease Control and Prevention (Korea). Infected MDCK cells were maintained in virus growth medium (EMEM) supplemented with 0.3% bovine serum albumin, 1% penicillin/streptomycin, and 0.5 µg/mL trypsin-TPCK (Sigma, Saint Louis, MO, USA). The virus titer was defined as the 50% tissue culture infectious dose (TCID50), according to the Reed–Muench endpoint method. All experiments involving virus-related work were performed in the Biosafety level 2 (BL2) facility using a biosafety cabinet.

#### 4.2.3. Cytotoxicity and Antiviral Assay

To determine the cytotoxicity of the TE extract, cell viability was determined by the water-soluble tetrazolium salt (WST) method using an EZ-Cytox kit (Daeil Lab Service, Seoul, South Korea) according to the manufacturer’s instructions. Briefly, MDCK cells were seeded in a 96-well plate at a density of 1 × 10^4^ cells/mL. After 24 h, cells were treated with a serial dilution of extract and incubated at 37 °C for 2 days. After this time the EZ-Cytox solution was added to each well and incubated in the dark for 2 h, followed by spectrophotometric measurements of optical density (OD) at 540 nm. The CC50 value of extract concentration inducing 50% cell death was calculated via regression analysis.

The antiviral activity of the TE extract against H1N1 infection was determined using the following four methods: pretreatment, attachment inhibition, co-treatment, and post-treatment assays. For all assays, MDCK cells were seeded in a 96-well plate and incubated for 24 h. In the pre-treatment assay, cells were first subjected to different concentrations of TE extract and incubated for 1 h at 37 °C. Next, samples were removed and infected with the virus (100 TCID50/well) for 1 h. In the attachment inhibition assay, the virus (100 TCID50/well) was pre-absorbed into cells before they were treated with different concentrations of TE extract for 1 h at 4 °C. The co-treatment assay involved incubating the cells with the virus (100 TCID50/well) and extract for 1 h at 37 °C. Lastly, for the post-treatment assay, cells were pre-incubated with the virus (100 TCID50/well) for 1 h at 37 °C, and then incubated for 48 h with various concentrations of the TE extract prepared in virus growth media. After 48 h, cell viability was measured using the WST method with the EZ-Cytox kit.

#### 4.2.4. Time-of-Addition Assay

To define the timeline of viral replication in which TE extract shows anti-influenza properties, a time-of-addition assay was performed. MDCK cells were seeded in a 96-well plate and incubated for 24 h. Next, cells were pre-incubated with the virus (100 TCID50/well) for 1 h at 37 °C. Later, the viral inoculum was removed, and cells were treated with 250 or 125 µg/mL of TE extract at different times post-infection. The experiment included treatment for 2–48 h, 4–48 h, 8–48 h, and 12–48 h post-infection. At the end of the experiment, cell viability was measured using the WST method with the EZ-Cytox kit.

#### 4.2.5. Plaque Reduction Assay

MDCK cells were seeded in a six-well plate and cultured for 24 h. Cells were infected with the virus at a dose of 50 plaque-forming units (PFUs) per well, and then incubated at 37 °C and 5% CO_2_ for 1 h. After removing the inoculum, the cells were treated for 1 h with different concentrations of TE extract. The mixture was then removed, and 3 mL of overlay medium containing 0.6% low-melting agar was added. The overlaid plates were incubated for 48 h at 37 °C and 5% CO_2_. Finally, the overlay medium was removed and the cells were subjected to crystal violet staining for plaque counting.

### 4.3. Statistical Analysis

The results are presented as the average of at least three replications. Differences between groups were analyzed using one-way analysis of variance and Dunnett’s test. Significance was set at *p* < 0.05. Data processing and statistical analyses were performed in GraphPad Prism 8.0.2 (GraphPad Software Inc., San Diego, CA, USA).

## 5. Conclusions

In conclusion, we identified lutein as a key active component in TE extract, largely responsible for its antiviral effects. Through an in silico study, we identified IL-6 as a hub target associated with influenza infection. Our research suggests that TE flower extract could be important as an alternative supplementation or treatment option for influenza in the future.

## Figures and Tables

**Figure 1 ijms-25-07065-f001:**
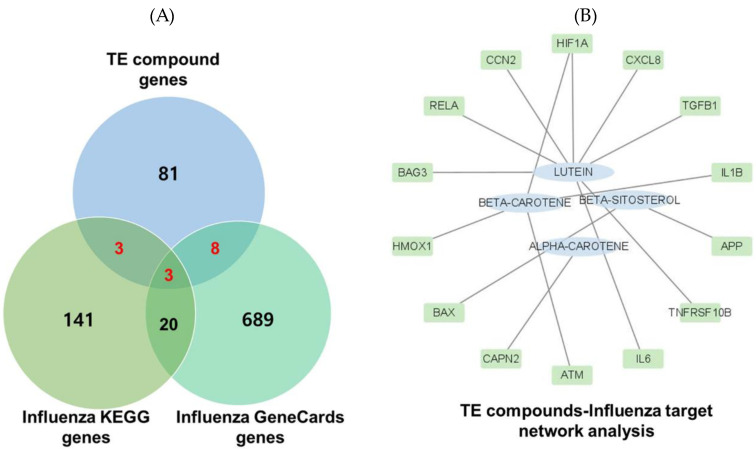
Target prediction. (**A**) Intersecting active targets. (**B**) Compound–disease target network analysis. Fourteen TE–influenza common targets were retrieved and constructed as a compound-disease target network.

**Figure 2 ijms-25-07065-f002:**
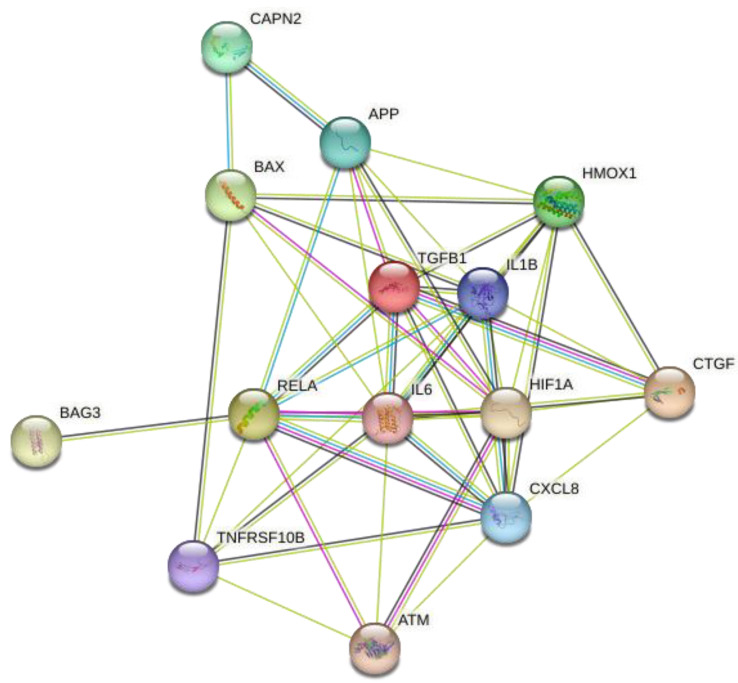
Target protein–protein interaction (PPI) network. Fourteen hub targets of TE and influenza were used as inputs for the STRING database. The network contained 14 nodes and 49 edges.

**Figure 3 ijms-25-07065-f003:**
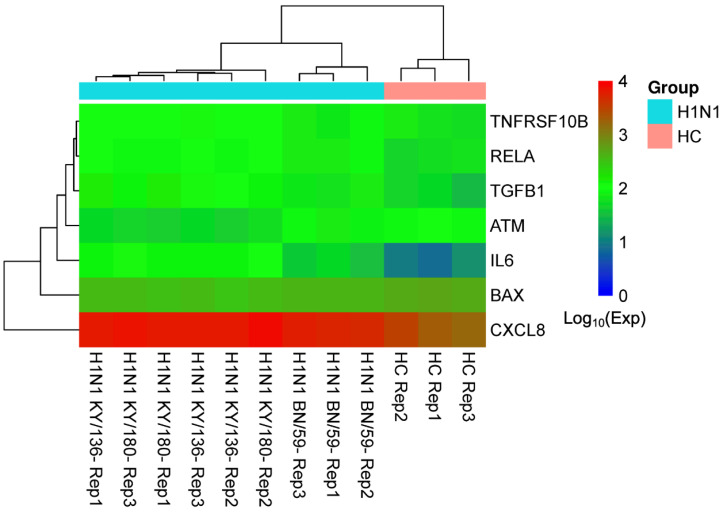
Gene difference heatmap of seven target genes. The Log_10_ expression levels of seven target genes (rows in the heatmap) are shown for nine H1N1 samples (columns with the prefix “H1N1” in the heatmap) and three healthy controls (columns with the prefix “HC” in the heatmap).

**Figure 4 ijms-25-07065-f004:**
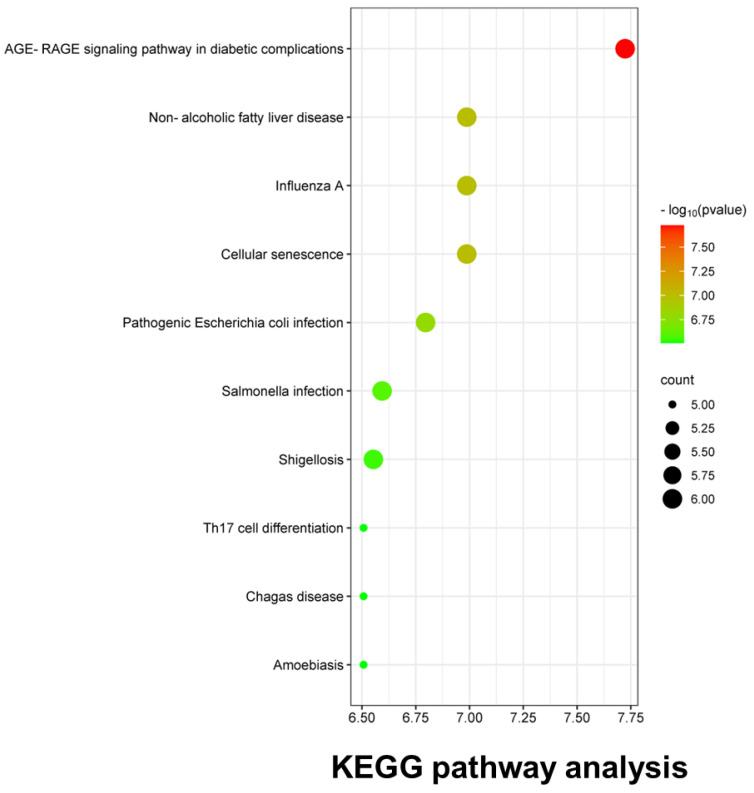
KEGG pathway enrichment analysis. TE and influenza−associated pathways and disease were screened and plotted in a bubble diagram according to the enriched gene counts.

**Figure 5 ijms-25-07065-f005:**
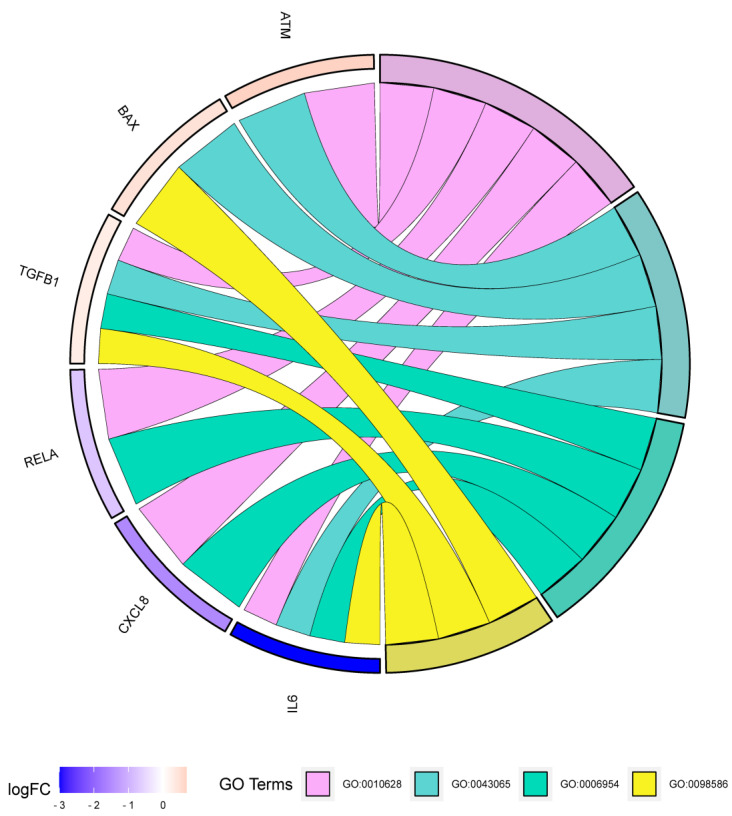
Gene ontology (GO) biological process (BP) analysis. The BP of the final screen targets was analyzed based on logFC, and six targets were enriched in a GO chord.

**Figure 6 ijms-25-07065-f006:**
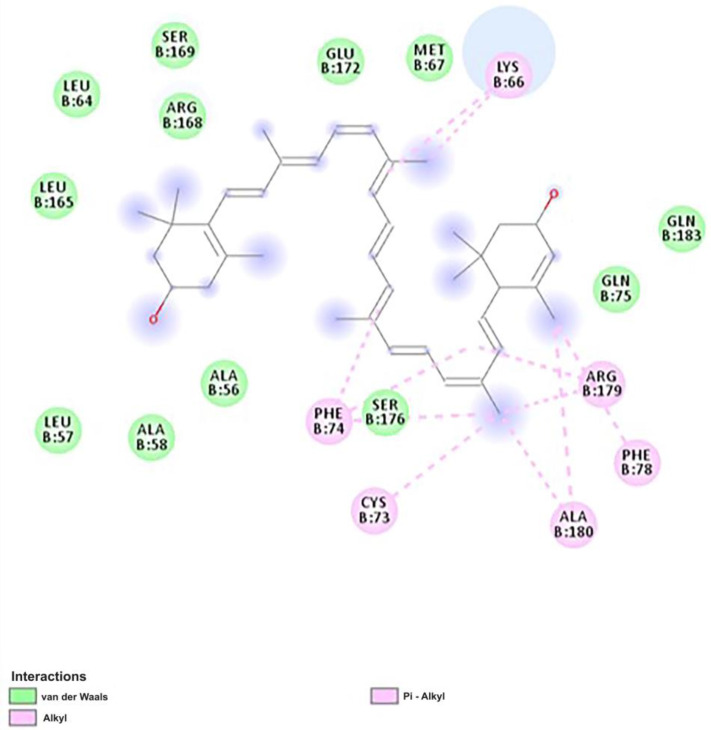
The 2D molecular docking of lutein and IL-6. The ligand–target interaction was visualized by using Discovery Studio Visualizer. The key bonds were alkyl bonds with LYS66, PHE74, CYS73, ALA180, and PHE78 as well as pi-alkyl bonds with PHE74 and ARG179.

**Figure 7 ijms-25-07065-f007:**
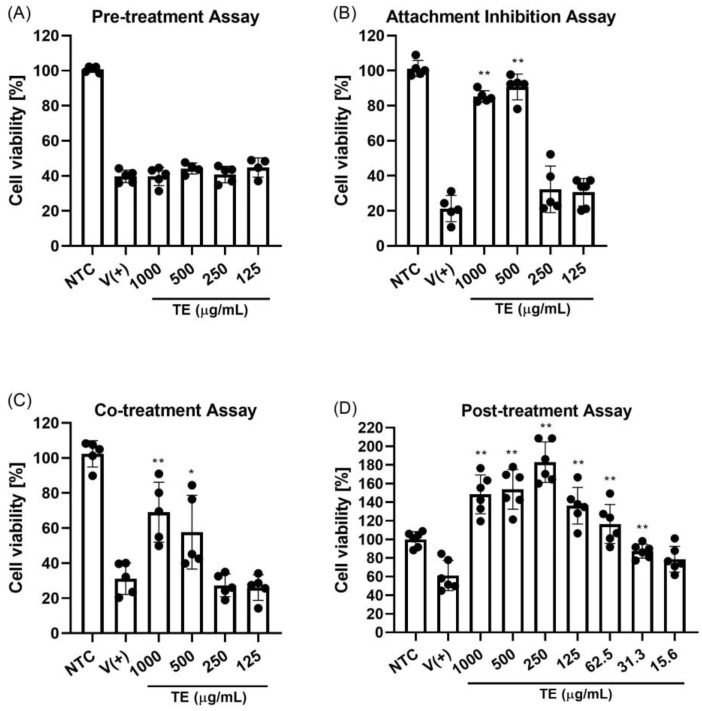
Antiviral properties of TE extract. (**A**) Pre-treatment assay: MDCK cells were treated with different concentrations of TE extract for 1 h at 37 °C, infected with influenza virus (100 50% tissue culture infectious dose [TCID50]/well), and incubated with viral inoculum for 1 h at 37 °C. (**B**) Influenza H1N1 virus attachment inhibition: MDCK cells were infected with influenza virus (100 TCID50/well) and co-treated with different concentrations of TE extract for 1 h at 4 °C. (**C**) Co-treatment assay: MDCK cells were co-incubated with influenza virus (100 TCID50/well) and different concentrations of TE extract for 1 h at 37 °C. (**D**) Post-treatment: MDCK cells were infected with influenza virus (100 TCID50/well) and incubated for 1 h at 37 °C. Virus was then removed, and cells were treated with different concentrations of TE extract. NTC: non-infected and non-treated control group; V (+): infected and non-treated group; ** *p* < 0.01, and * *p* < 0.05.

**Figure 8 ijms-25-07065-f008:**
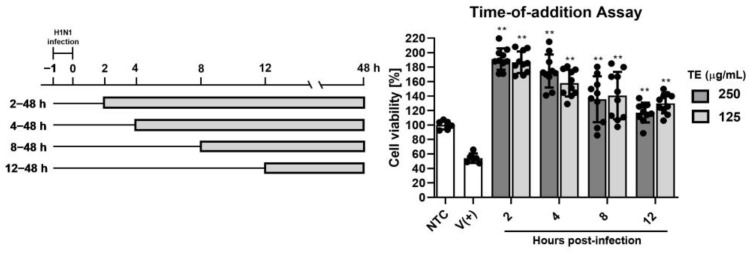
Time-of-addition assay: MDCK cells were infected with influenza virus (100 TCID50/well) and incubated for 1 h at 37 °C (−1 to 0 h). Virus was removed, and cells were treated with 250 or 125 µg/mL of TE extract starting 2, 4, 8, or 12 h after infection. NTC: non-infected and non-treated control group; V (+): infected and non-treated group; ** *p* < 0.01 (compared with virus-only control).

**Figure 9 ijms-25-07065-f009:**
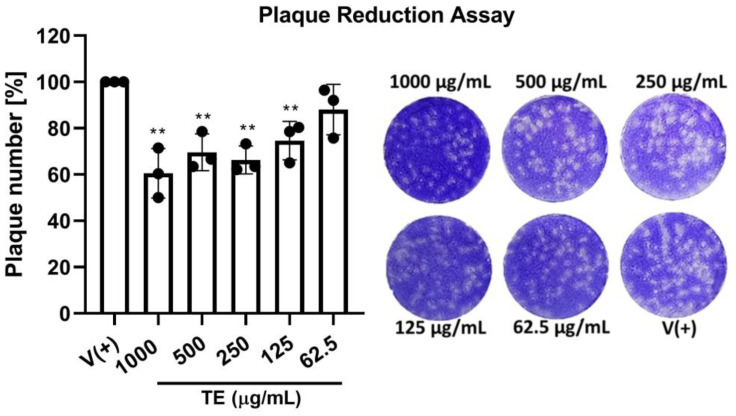
Plaque reduction assay. MDCK cells were infected with influenza H1N1 virus (50 plaque-forming units [PFUs]/well) and incubated for 1 h at 37 °C. Next, cells were treated with different concentrations of TE extract and incubated in the same conditions for 1 h. Compound mixtures were then removed, and 3 mL of overlay medium was added to each well. After 2 d, cells were stained with a crystal violet solution and plaques were calculated. Plaque reduction was determined via comparison with untreated infected wells (*n* = 3). ** *p* < 0.01 (compared with virus-only control).

**Table 1 ijms-25-07065-t001:** Active compounds in the *Tagetes erecta* (TE) extract.

Compound	Classification	MW	OB	DL
Alpha-carotene	Carotenoid	536.96	34.51	0.58
Beta-carotene	Carotenoid	536.96	37.18	0.58
Beta-sitosterol	Phytosterol	414.79	36.91	0.75
Campesterol	Phytosterol	400.76	37.58	0.71
Lutein	Carotenoid	568.96	22.59	0.55
Stigmasterol	Phytosterol	412.77	43.83	0.76

MW: molecular weight; OB: oral bioavailability; DL: drug-likeness.

**Table 2 ijms-25-07065-t002:** Intersection target prediction.

Target	Description	Degree
IL-6	Interleukin-6	11
HIF1A	Hypoxia-inducible factor 1 subunit alpha	10
IL-1β	Interleukin-1 beta	10
CXCL8	Interleukin-8	9
RELA	RELA proto-oncogene, NF-κB subunit	9
HMOX1	Heme oxygenase 1	8
TGFB1	Transforming growth factor beta 1	8
APP	Amyloid beta precursor protein	7
BAX	Apoptosis regulator BAX	6
CTGF	Cellular communication network factor 2	6
TNFRSF10B	Tumor necrosis factor receptor superfamily member 10B	6
ATM	ATM serine/threonine kinase	5
CAPN2	Calpain 2	2
BAG3	BAG cochaperone 3	1

**Table 3 ijms-25-07065-t003:** Binding affinities of the active compounds of the TE extract with influenza-associated genes.

Compound	Binding Affinity (kcal/mol)		
	IL-6	TGFB1	CXCL8
Lutein	–8.6	–7.3	–7.3
Beta-carotene	–8.2	–7.6	–7.6

**Table 4 ijms-25-07065-t004:** CC_50_, IC_50,_ and SI values of the TE extract.

CC_50_ (µg/mL)	IC_50_ (µg/mL)	SI
>2000	76 ± 18	>30

CC_50_: mean cytotoxicity concentration of 50% compared with the control. IC_50_: mean half-maximal inhibitory concentration determined with WST method in the post-treatment assay. SI: selectivity index (CC_50_/IC_50_).

## Data Availability

Requests for data sharing should be addressed to the corresponding author.

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
