# Peer review of "Anti-Influenza A Potential of Tagetes erecta Linn. Extract Based on Bioinformatics Analysis and In Vitro Assays"

_ijms, 2024, doi:10.3390/ijms25137065_

Round 1

Reviewer 1 Report

Comments and Suggestions for Authors

Dear Editor

Many thanks for considering me a potential reviewer for the said article entitle; Anti-influenza A potential of Tagetes erecta Linn. extract based on bioinformatics analysis and in vitro assays. This article is well structured and written, however, here are some major comments and minor corrections (suggestions) that must be taken into consideration before the onward steps.

My observations are as follow;

Major Comments

1.     Tagetes erecta and other scientific names should be italicized, throughout the MS,

2.     Molecular docking analysis…Line 274 ‘site of targets……..Please expend which parameters were considered step by step….Ligand-Target’,

3.     Abstract; A), Please reduce the material and method in abstract section, since its already repeated in RESULTS and MATERIAL AND METHOD. B), Please add some results about the docking studies (what you observed) and relate it to in vitro results (possible confirmation).

4.     Line 206-208; a) Herein, we conducted molecular docking to assess target binding of the two main compounds in the TE extract. We found that lutein binds more strongly than beta-carotene to IL-6. Is this is discussion?. I am strongly recommending, please add information regarding the binding what you observed/find are in agreement and/or in contradiction?. B) You had mentioned in the abstract ‘We identified lutein and beta-carotene as the main active compounds in TE extract, and IL-6 as an important target associated with influenza by using data mining and bioinformatics. But in the results you just showed the docking results of IL-6 and lutein, why not beta-carotene?, If the beta-carotene was not interesting then correct your statement in the abstract and/or add biding results of beta-carotene with IL-6 too. C), Just a curiosity, what about natural inhibitor of IL-6? Did you consider for the comparison? If no then how you can claim your compounds are better….

Minor comments  

1.     The citation style [1], please correct,

2.     P < 0.001…. p< 0.05 should be italicized throughout the MS,

3.     Cite these information ‘Ethanol was used as it is safe for infused edibles and provides consistent results.’ Line 280,

4.     Line-321, Significance was set at P < 0.05….is this is right claim?, I think no, because you had mentioned different levels in each graphs description/legends for example Fig.8,

5.     Improve Figure 6 quality,

6.     Line 22, please change this sentence, In vitro experimental results revealed that the TE extract exerted therapeutic rather than protective effects.

7.     Please change the caption (µg/ml to µg/mL) in the figure 7 and 8.

8.     Please add one column Table 1 regarding the classification of the compounds, it will make the study more informative.

Comments on the Quality of English Language

Dear Authors/Editor 

Thanks for the consideration.

Besides, the English grammer/mistakes, I noticed that material and methods are repeated in the results too, I will suggest to proofread this MS by a senor Prof. Professional in the field. 

Thanks

Author Response

Thank you and we appreciate all your comments.

Reviewer 2 Report

Comments and Suggestions for Authors

Thank you for the opportunity to review the article entitled "Anti-influenza A potential of Tagetes erecta Linn. extract based on bioinformatics analysis and in vitro assays"

The manuscript represents an original reserach based on computational simulation of potential interactions of bioactive molecules in Tagetes erecta flower extract and cell factors participating in the influenza viral infection.

The paper has clear title, the study design is rational and logically ordered including first in silico analysis followed by standard classical virological assays in vitro. Materials and methods are well-described, results are supported by clear figures, graphs and pictures.Discussion involves intensive comparison with other data in the same field. The reserach is valuable and is an excellent fundament for a more detailed and profound biological anlaysis to explore and explain the potentials of the extract. In my opinion, some essential additional  molecular and virological techniques are needed to support the results from the mathematical modelling. Here are my comments and questions:

1. All the Latin names and phrases are supposed to be in italics.

2.  Plant extract composition - it is established, I guess. If so, information should be provided.

3. There is no any comment on doses used in the antiviral assays. How were the doses selected if there is no cytotoxicity test? No data of CC50, IC50 and SI calculated. That is essential in the antiviral studies.

4. I could not understand exactly what is the essential difference between viral adsorption inhibition and co-treatment assay. Altough the four types of treatmet give relative orientation if the extract works on the early or late stages of the replication (late, in th the case), it appears to me that Time-of- addition test would be also quite informative.

5.  Interactions with the factors of inflammation of lutein and beta carotene shown by mathematical modelling should be confirmed by biological tests in vitro to support the conclusion for the inclusion of Tagetes extract in the therapy of influenza.

6. Two antiviral tests in vitro are conducted using MDCK cells. Although MDCK is a standard and well-established cell culture model for influenza in vitro antiviral drugs studies I find it appropriate to use human cell line, such as A549, Calu-3, primary bronchial epithelial cells or other having in mind that host-cell factors are affected. 

7. I completely agree with the authors' statement that the extract should be investigated for its antioxidant capacities and that the antioxidants play an essential role in the complex therapy of influenza.

8.  Within the expression Line 23"In vitro exper imental results revealed that the extract exerted therapeutic rather than protective effects" I find as a more appropriate and precise "prophylactic" or preventive"  to replace protective since it considers course of application.

Regardless of the above mentioned remarks I find that the paper could be published after some corrections are introduced.

Author Response

(The authors gave the same response as above.)

Round 2

Reviewer 1 Report

Comments and Suggestions for Authors

Dear Authors,

Thank you very much for the  consideration and respond to each and every comment. The article is much and more refined. 

Regards,

Comments on the Quality of English Language

Dear Authors/Editors,

Many thanks for the consideration and updates.

The article is much more refined, however, I am not happy with English. I will suggest please do English editing by a professional in the field and/or a senior professor.

Thanks 

Reviewer 2 Report

Comments and Suggestions for Authors

Dear Authors,

Thank you very much for responses, additional information included and corrections made. I think, now the work stands much more convincing and complete. I am also grateful for giving your results on antioxidant properties - that would be a really promising candidate for a broad-spectrum herbal antiinfluenza therapeutic. I can recommend the article to be accepted in this form for publication.

Best regards